# Modeling the Impact of Fiscal Decentralization on Energy Poverty: Do Energy Efficiency and Technological Innovation Matter?

**DOI:** 10.3390/ijerph20054360

**Published:** 2023-02-28

**Authors:** Yaru Wang, Guitao Qiao, Mahmood Ahmad, Dan Yang

**Affiliations:** 1School of Economics and Management, Weifang University, Weifang 261061, China; 2Business School, Shandong University of Technology, Zibo 255000, China

**Keywords:** fiscal decentralization, energy poverty, energy efficiency, technological innovation, China

## Abstract

As an important factor affecting economic and social development, energy poverty (EP) has received widespread concern, and many countries have actively proposed policies to eliminate energy poverty. The purpose of this paper is to clarify the current situation of energy poverty in China, explore the factors that affect energy poverty, find sustainable and effective approaches to alleviate energy poverty, and provide empirical evidence for eliminating energy poverty. This research investigates the effect of fiscal decentralization (FD), industrial structure upgrading (ISU), energy efficiency (EE), and technological innovation (TI), as well as urbanization (URB) on energy poverty using a balanced dataset of 30 provinces in China from 2004 to 2017. The empirical outcomes revealed that fiscal decentralization, industrial upgrading, energy efficiency, and technological innovation significantly reduce energy poverty. Moreover, urbanization is positively and significantly correlated with energy poverty. The outcomes further revealed that fiscal decentralization significantly increases the residents’ access to clean energy and drives energy management agencies and infrastructure. In addition, the heterogeneity analysis results indicate that the effect of fiscal decentralization in reducing energy poverty is greater in regions with high economic development. Finally, mediation analysis denotes that fiscal decentralization indirectly reduces energy poverty by promoting technological innovation and energy efficiency. Finally, based on the results, policy suggestions for eradicating energy poverty are proposed from the perspective of implementing targeted energy alleviation policies reasonably dividing the rights and responsibilities of local and central governments and encouraging scientific and technological innovation.

## 1. Introduction

In 2014, the United Nations put forward the Sustainable Development Goal (SDG) to “eradicate poverty in all its forms,” and energy poverty is commonly regarded to be a manifestation of poverty and a determinant of enduring poverty [1], which has attracted widespread attention. Energy poverty restricts not only national economic development and social progress [2], but also causes ecological damage and pollution by relying on the combustion of traditional biomass energy [3]. Thus, energy poverty has been identified as a major global problem because of its adverse effects on social development, economic growth, and climate change [4]. The 2030 agenda for sustainable development sets out goals to ensure access to affordable, reliable, and sustainable modern energy for all, to double the global energy efficiency, and to promote research and technology for access to clean energy.

Energy poverty emerged initially in the fuel access movement in the UK, which highlighted the inability of households to afford necessary energy services. As economies have developed and time has progressed, energy poverty has come to represent different characteristics in developed and developing countries [5] and has been given new definitions. Boardman [6] defines energy poverty as manifested in the inability to pay for adequate energy services, and Hills [7] defines it as low income and high cost. IEA [8] put forward the definition of energy poverty from a perspective that is more in line with developing countries, which rely on traditional biofuel for life activities without the ability to obtain and use clean energy, for example, electricity. Based on the Chinese context, Wang et al. [9] argue that energy poverty must address not only survival needs but also security and development needs, and they construct a comprehensive evaluation index system comprising four dimensions to measure energy poverty.

Energy poverty has emerged as a prominent topic of interest among scholars, who have extensively investigated its economic consequences. Firstly, energy poverty hinders economic poverty eradication and economic development [2]. Amin et al. [1] argue that energy poverty and poverty are essentially homologous, that poverty is an important aspect of energy poverty and that the two can be reduced at the same time. Secondly, energy poverty is often associated with numerous social issues, such as education and health [10,11]. Energy deprivation affects education and negatively affects the average school year of households [12]. Households experiencing energy poverty often rely on biofuels (such as wood, coal, dung, and waste) for their household energy needs. However, the combustion process of these fuels is often characterized by low efficiency and poor ventilation, which can have harmful effects on respiratory health. [13]. Furthermore, some scholars argue that energy poverty affects environmental quality [9,14].

With a large population, limited resources, and uneven regional development, China has an obvious problem of low energy use and poor energy use structure. Particularly, in rural areas, more households use traditional biomass as fuel, which affects the health of rural residents as well as ecological degradation [13,14]. In response to the energy problems of China, the Chinese government has developed policies to relieve energy poverty, for instance, the utilization of photovoltaic projects to help alleviate poverty; the “1 + N” policy system proposed by the State Council’s Poverty Alleviation Office, which is one document plus a number of supporting policy issue documents. At present, China has solved the problem of electricity difficulty after years of hard work. The effectiveness of energy poverty alleviation is evident and has been affirmed by the Chinese government and people. However, this does not mean the end of poverty alleviation work. On the contrary, there is still a long way to go to consolidate the achievements of poverty alleviation, and higher requirements for energy poverty alleviation are put forward.

To address energy poverty, numerous scholars have explored various possible influencing factors. For example, economic development [15,16], electricity penetration [17], renewable and clean energy [18], and energy efficiency [13,19,20] all have a positive effect for eradicating energy poverty. Fiscal decentralization refers to the process by which the central government delegates some of its fiscal power to local governments to enhance their autonomy. This can significantly influence local energy utilization. Many scholars have reached an agreement on the conclusion that FD has an impact on energy, but there are contradictions on the impact. Zhang et al. [21] considered two aspects of the influence of FD on the environment, namely “race to the top” and “race to the bottom”, to examine the nexus between FD and renewable energy. The phenomenon of “race to the top” will improve environmental quality, and this result depends on renewable energy. The “race to the bottom” is exactly the converse. In pursuit of interests, local governments attract more investment in economic activities with a cost of environmental damage, which is raised to a massive use of non-renewable resources, negatively affecting renewable energy consumption [22]. Elheddad et al. [23] proposed an inverted U-shaped nexus between FD and energy consumption. Coincidentally, Kassouri [24] used threshold and quantile regression techniques to test the nexus between FD and renewable energy R&D, and confirmed that the relationship between the two is nonlinear and heterogeneity in the types of FD. At present, the research on the relationship between fiscal decentralization and energy poverty is not rich, and whether fiscal decentralization can play a role in alleviating energy poverty has not been fully revealed, which provides a new research opportunity for this paper to further explore.

Based on the above gaps, this paper examines the impact of FD on energy poverty in a sample of 30 Chinese provinces from 2004 to 2017. This paper studies the impact of fiscal decentralization, industrial upgrading, energy efficiency, technological innovation, and urbanization development on energy poverty. The results show that there is a negative and significant correlation between fiscal decentralization, industrial upgrading, energy efficiency, technological innovation, and energy poverty, while the acceleration of urbanization will increase energy poverty. The fiscal decentralization system will improve the right and autonomy of local governments to deal with energy issues, and at the same time, with a better understanding of the local development situation, local governments will have a more significant effect on addressing energy issues [24]. Overall, the upgrading of industrial structures and energy efficiency as well as technological innovation will improve the efficiency of resource allocation, at the same time, promote the popularization and use of modern and efficient clean energy [25]; The acceleration of urbanization will increase the consumption of fossil energy [26] and hinder the solution of energy poverty. In addition, the heterogeneity test and intermediary test are also applied to further explore the impact of fiscal decentralization on energy poverty under different circumstances and internal mechanisms.

The research contributions and significance of this paper are mainly shown in the following aspects. Firstly, from the perspective of fiscal decentralization, this paper provides new ideas for alleviating energy poverty and enriches the existing literature. This paper analyzes the possible causes of energy poverty, and the possible ways to solve energy poverty from the perspective of government fiscal power. At the same time, it analyzes the potential influencing factors, and enriches the relevant literature in the field of energy poverty. Secondly, based on China’s background, this paper explores ways to alleviate energy poverty, which plays a certain role in promoting China’s energy poverty alleviation work. As one of the most influential developing countries in the world, China has a large population base and unbalanced regional development, which will undoubtedly lead to more severe energy poverty, particularly in the isolated region. Hence, there is an urgent need to study the effect of fiscal decentralization on energy poverty under the background of China. Finally, the present paper explores the differences in the relationship between fiscal decentralization and energy poverty under different circumstances, providing empirical evidence for proposing more specific energy poverty alleviation policies. Through the heterogeneity test, the energy poverty is subdivided into sub-indicators to explore the impact of fiscal decentralization separately, and the differences of the impact of fiscal decentralization on energy poverty under different economic development conditions are explored. In addition, the impact mechanism of fiscal decentralization on energy poverty is analyzed to provide more detailed focus for fiscal decentralization to play its role and provide a basis for more targeted policy suggestions.

The remainder of the present study is structured as below. Section 2 presents the theoretical framework, data, and econometric methods. Section 3 denotes the estimation methods and explain the empirical results. Section 4 is based on further analysis to explore the internal impact mechanism of fiscal decentralization on energy poverty. A discussion on the results is given in Section 5. Section 6 discusses the conclusions and policy implications.

## 2. Theoretical Framework, and Data

### 2.1. Theoretical Framework

The nexus between FD and EP, and its mechanism is discussed below. Firstly, the theory of fiscal federalism suggests that decentralization of fiscal power can promote efficient and effective public service delivery at the local level by creating incentives for local governments to improve their own revenue base and expenditure decisions [27]. This theory implies that fiscal decentralization may have a positive impact on energy poverty alleviation by empowering local governments to develop and implement policies and programs that target the needs of energy-poor households. FD implies greater autonomy for local governments, which may promote local economic development in the form of attracting circulating capital and introducing foreign investment [28,29,30], thereby increasing the income of the residents. As a result, residents’ opportunities and capabilities to validly access modern energy services will increase, which will help reduce energy poverty. Hence based on the above argument, FD is expected to have a negative effect on energy poverty $\left(\alpha_{2}=\frac{\partial EP}{\partial FD}<0\right)$. Moreover, FD results in the phenomenon of “race to the top” among local governments [31,32], and local governments will be more attentive to environmental protection issues. Secondly, structural transformation in the context of economic development theory suggests that economic development is driven by a shift from low-productivity agricultural activities to high-productivity industrial activities. This transformation is accompanied by a change in the structure of the economy, as resources are reallocated towards more productive sectors. In the context of energy poverty, industrial upgrading can play a critical role in reducing energy poverty. As the industrial sector becomes more productive, it creates jobs and raises incomes, which can increase access to energy services. Moreover, industrial upgrading can lead to greater energy efficiency and the adoption of cleaner technologies, which can reduce the reliance on fossil fuels and mitigate the negative environmental impacts associated with energy poverty [20,33]. Thus, it is expected that industrial structural transformation has negative impact on energy poverty $\left(\alpha_{3}=\frac{\partial EP}{\partial ISU}<0\right)$.

Theoretically, energy efficiency can help reduce energy poverty by reducing the overall energy demand and therefore, the cost of energy consumption for households [20]. This reduction in energy demand and cost can make energy services more affordable and accessible to low-income households, thereby improving their energy access and reducing their vulnerability to energy poverty. Hence, energy efficiency is projected to have a negative impact on the energy poverty $\left(\alpha_{4}=\frac{\partial EP}{\partial EE}<0\right)$. Technologies can improve the efficiency of energy production, distribution, and consumption, reducing the cost of energy and making it more accessible to low-income households [34]. For instance, the development of renewable energy technologies such as solar panels and wind turbines can provide a cheaper and more sustainable source of energy for households that are unable to afford traditional forms of energy. Secondly, technological innovations can improve energy access by expanding the reach of energy infrastructure, such as through the use of micro-grids or smart grid technologies. Thirdly, new technologies can enable households to use energy more efficiently, reducing their overall energy consumption and associated costs. Overall, technological innovation has the potential to play a critical role in addressing energy poverty by increasing energy efficiency, reducing costs, and improving access to affordable and sustainable energy sources. Hence, it is predicted that technological innovation lessens energy poverty $\left(\alpha_{5}=\frac{\partial EP}{\partial TI}<0\right)$. The acceleration of urbanization will be accompanied by a large amount of energy consumption [35]. On the one hand, the expansion of urban scale will bring the growth of the economy and increase the amount of infrastructure. Whether it is built or put into use, it will consume a lot of energy service. On the other hand, urbanization will increase the population, especially promote the migration of the rural labor force to cities. At the same time, the energy consumption level of urban residents is usually higher, so the energy demand will be increased. Of course, the impact of urbanization on energy poverty may be related to the development stage, but in terms of the current urbanization process in China, the development of energy-intensive industries is likely to increase energy poverty $\left(\alpha_{6}=\frac{\partial EP}{\partial URB}>0\right)$. This paper process involves the natural logarithms on all variables to cope with possible heteroskedasticity, as well as estimation bias due to unit inconsistency. In addition, given the dynamics and potential time-lag effect of EP, this research uses a dynamic panel model that incorporates the dependent variables lagged by one period, as shown in Model 1.
(1)$$\begin{array}{c} {\mathrm{LnEP}}_{\mathrm{it}}=\alpha_{0}+\alpha_{1}{\mathrm{LnEP}}_{i,t-1}+\alpha_{2}{\mathrm{LnFD}}_{i,t}+\alpha_{3}{\mathrm{LnISU}}_{i,t}+\alpha_{4}{\mathrm{LnEE}}_{\mathrm{it}}+\alpha_{5}{\mathrm{LnTI}}_{\mathrm{it}}+\\ \alpha_{6}{\mathrm{LnURB}}_{i,t}+\varepsilon_{\mathrm{it}} \end{array}$$where EP is energy poverty, ${\mathrm{EP}}_{i,t-1}$ denotes the lag period of EP, and FD is the independent variable, representing fiscal decentralization. ISU, EE, TI, and URB respectively represent industrial upgrading, energy efficiency, technological innovation, and urbanization development. i represents the province, t denotes the year, and ε is a random disturbance term.

Under the pressure of “race to the top”, local governments will raise awareness of environmental protection and increase investment in environmental protection. An important factor contributing to EP is the availability of energy. With the improvement in energy efficiency, energy demand will gradually decrease, which will greatly help EP alleviation. Li et al. [20] and Boemi and Papadopoulos [36] suggest that energy efficiency and EP are positively related. China has established five categories of energy efficiency standards to raise the energy efficiency of household appliances and cut down energy consumption. Hence, the joint effect of FD and energy efficiency is supposed to have a negative impact on energy poverty $\left(\beta_{6}=\frac{\partial EP}{\partial \left(FD*EE\right)}<0\right)$. Zameer et al. [37] suggest that technological innovation improves productivity and poverty alleviation efficiency. Technological innovation can improve energy utilization efficiency [38], drive a shift in the energy structure from fossil fuel to cleaner energy [39,40], and push forward the development of an efficient energy market [41]. The development of technological innovation has brought new technologies, maximized the use of resources, and promoted the use of new energy, which is conducive to reducing the consumption of fossil fuels [42,43] and eliminating energy poverty. Therefore, the higher the level of energy efficiency and technological innovation, the more adequate conditions local governments will face, and the greater the contribution to reducing energy poverty. This study hypothesizes that the effect of FD on EP strengthens as technological innovation improvements $\left(\gamma_{6}=\frac{\partial EP}{\partial \left(FD*TI\right)}<0\right)$. In order to examine the moderation effects of technological innovation and energy efficiency on energy poverty, this paper introduces the interaction terms lnFD * lnEE and lnFD *lnTI into Model 1 respectively, as shown in Model 2 and Model 3. Considering the problem of multicollinearity, lnEE or lnTI were not included in the model while adding the interaction terms.
(2)$${\mathrm{LnEP}}_{\mathrm{it}}=\beta_{0}+\beta_{1}{\mathrm{LnEP}}_{i,t-1}+\beta_{2}{\mathrm{LnFD}}_{\mathrm{it}}+\beta_{3}{\mathrm{LnISU}}_{\mathrm{it}}+\beta_{4}{\mathrm{LnTI}}_{\mathrm{it}}+\beta_{5}{\mathrm{LnURB}}_{i,t}+\;\beta_{6}{\mathrm{LnFD}}_{\mathrm{it}}{*\mathrm{LnEE}}_{\mathrm{it}}+\varepsilon_{\mathrm{it}}$$
(3)$${\mathrm{LnEP}}_{\mathrm{it}}=\gamma_{0}+\gamma_{1}{\mathrm{LnEP}}_{i,t-1}+\gamma_{2}{\mathrm{LnFD}}_{\mathrm{it}}+\gamma_{3}{\mathrm{LnISU}}_{\mathrm{it}}+\gamma_{4}{\mathrm{LnEE}}_{\mathrm{it}}+\gamma_{5}{\mathrm{LnURB}}_{i,t}+\;\gamma_{6}{\mathrm{LnFD}}_{\mathrm{it}}{*\mathrm{LnTI}}_{\mathrm{it}}+\varepsilon_{\mathrm{it}}$$

### 2.2. Variables Measure and Data Sources

In order to study the effect of FD on EP, a balanced panel of 30 Chinese provinces from 2004–2017 is used in this study. Considering the data availability, the sample data of Tibet, Hong Kong, Macau, and Taiwan are not covered. The starting period of 2004 is based on FD data and the ending period of 2017 is linked with data availability for EP.

The current study used energy poverty as the dependent variable. At present, no uniform measure of energy poverty has been derived, and this paper draws on Wang et al. [9] to classify EP into four dimensions: energy service availability (ESA), energy consumption cleanliness (ECC), energy management integrity (EMC), household energy affordability, and energy efficiency (EAE), calculated following the improved entropy method (IEM) by 17 indicators. Data were obtained from the work of Dong et al. [5].

Fiscal decentralization was used in this paper as an independent variable, which is measured by the ratio of the provincial per capita local fiscal expenditure to the per capita central fiscal expenditure, referring to Cheng et al. [44]. This indicator measures the division of responsibilities. The level of expenditure undertaken by the local government represents the involvement in the local economy. The data come from the China Fiscal Statistical Yearbook.

Furthermore, the data of industrial structure (ISU), technological innovation (TI), and urbanization development (URB) were obtained from the China Statistics Yearbook. The data of energy efficiency (EE) are acquired from the China Energy Statistics Yearbook and China Statistics Yearbook. See Appendix A Table A1 for specific indicators and sources. The descriptive statistics of each variable are displayed in Table 1 below.

## 3. Estimation Methods and Empirical Findings

### 3.1. Spatial-Temporal Analysis

This paper presents a spatial-temporal analysis of EP and FD to explore the temporal trends in these two variables as well as inter-regional differences. Table 2 demonstrates the EP differences between regions. The largest average EP indicator is found in the western region and the smallest in the eastern area, indicating that the western area is most constrained by EP. The western region is mostly underdeveloped areas, compared with the eastern coastal areas with higher economic development levels, residents’ income levels in the western region are lower, energy infrastructure construction is slower, and the popularization of modern clean energy is insufficient. These factors ultimately result in a limited capacity and access to energy. Figure 1 demonstrates that in the eastern region, especially the coastal areas, shows a low level of energy poverty, while the northwest and northeast regions are troubled by energy poverty, which presents the pattern of “low in the east and high in the west.” It is noteworthy that Shanxi, as a central region, has a high level of energy poverty. This may be due to the fact that, as a major coal province, Shanxi has a homogeneous energy use structure and serious pollution emissions. The abundance of coal resources has in turn become a barrier to reducing energy poverty.

Figure 2 displays a graph of the trends for average EP and average FD over time. It can be seen that from 2004 to 2017, the degree of fiscal decentralization showed an upward trend, while the level of energy poverty continued to decline. According to the time change trend for the four sub-indicators of energy poverty, it can be seen that the decrease in ESA (access to clean energy) is the main force leading to the decrease in EP. With the increasing living standards, residents’ access to clean energy has increased, and that improves energy poverty eradication.

### 3.2. Preliminary Check

#### 3.2.1. Cross-Sectional Dependence Test

Taking into account the possible correlation among provinces due to spillovers of policies and technologies, the present study applied Breusch and Pagan (1980), Pesaran (2004), and Friedman (1937) approaches for cross-sectional dependence checking. The outcomes are displayed in Table 3. The three tests all indicate the existence of cross-sectional dependence between provinces. This means that there is a spillover effect between provinces, that is, policies, energy facilities, and technological levels in one province will have an impact on its neighboring provinces.

#### 3.2.2. Unit Root Test

In light of the issue of cross-sectional correlation, this research applied Dickey–Fuller’s cross-sectionally augmented (CADF) method and Pesaran’s (2007) cross-sectionally augmented IPS (CIPS) method to investigate the order of integration level of the model parameters. CADF and CIPS are second-generation unit root tests, and both are robust to cross-sectional dependency. The consequences are listed in Table 4, where the second and third columns denote the CIPS statistics for the level and first-difference forms of the variables, while the fourth to fifth columns represent the CADF statistics, respectively. The outcomes demonstrated that a mixed order of integration and most the variables are stationary after their first difference.

### 3.3. Baseline Regression

This study used FGLS and differential GMM for benchmark regression to control for the effect of heteroskedasticity and autocorrelation on the estimated results. The outcomes are displayed in Table 5, where the first three columns of the outcomes indicate estimation of models 1 to 3 with FGLS and the last three columns are estimated with GMM. According to the results of model 1, the significance as well as the direction of the variables are approximately the same between the two estimation methods, except for the lnURB coefficient, which is not significant in the FGLS method. These results illustrate that our model is more suitable for diff-GMM. The current research, therefore, holds that the diff-GMM is likely to be more accurate and valid for the model set out in this paper, so the results of the diff-GMM are mainly analyzed here. The results of Arellano–Bond (A-B)’s statistic AR(1), and AR(2) demonstrate that the first-order difference of the error term is significant, while the second-order difference is not significant, and the Sargan statistic shows that the instrumental variables are effective.

The results in Model 1 show that FD is negatively related to EP. Its elasticity coefficient is −0.273, which means that growth in FD by one unit will reduce energy poverty by 0.273%. The Chinese government has released numerous new energy policies and issued several guidance plans to drive the development of new energy. For example, the State Council has issued several documents to promote the development of rural photovoltaics and wind power, improve the pluralistic utilization of rural biomass energy, and speed up the construction of a rural clean energy utilization system based on renewable energy. In addition, a special fund has been established to provide subsidies, tax relief, and a large amount of scientific research investment in talent training.

The relationships among ISU, EE, and TI are negative and significant, while the URB and EP relationships are significantly positively correlated. These results are the same as our expectations. The upgrading of industrial structure can heighten the efficiency of resource distribution, promote technological development, and accelerate the process of eliminating EP. The improvement in energy efficiency will help lower the cost of energy consumption, increase the supply of energy-efficient products and services, and promote the popularization of modern and efficient energy. Similarly, the development of technological innovation helps to drive the shift of energy consumption structure and the development of clean energy to replace traditional high-polluting energy. In the process of accelerating urbanization, it is accompanied by the consumption of abundant fossil energy, which is adverse to the elimination of EP. The relationship between control variables and EP is roughly the same as the consequences of Dong et al. [45].

For the interaction terms, lnFD *lnTI and lnFD *lnEE were significantly negatively correlated with EP and the elasticity coefficients in the diff-GMM regression were −0.014 and −0.046, respectively. This suggests that EE and TI play a moderate role between FD and EP. The advantages of EE and TI in energy poverty alleviation make local governments more comfortable in dealing with energy issues.

### 3.4. Heterogeneous Analysis

In this part, this paper explores the effect of FD on sub-indicators of EP and the impact of FD on EP at different economic development levels. Using diff-GMM, the empirical outcomes are displayed in Table 6. The elasticity coefficient of the relationship between lnFD and lnESA is −0.085, which is significantly correlated at the 5% level. The elasticity coefficient of the relationship between lnFD and lnESA was −0.085, which was significantly correlated at the 5% level. ESA denotes energy service availability; the result illustrates that raised fiscal decentralization increases residents’ access to clean energy. Local governments are more familiar with the status quo, characteristics, patterns, and demands of regional economic development. High decentralization is an advantage for local governments to better formulate policies based on their local advantages and characteristics. This is not only conducive to the improvement in the quality of economic development but also drives the level of residents’ disposable income, which is very effective for energy poverty alleviation. ECC represents the cleanliness of energy consumption, and the relationship between lnFD and lnECC is positively and significantly correlated. The possible reason for this result may be that local governments pursue rapid economic development with large consumption of fossil fuels, while clean energy has not been effectively utilized. EMC and EAE stand for energy management integrity and the affordability and efficiency of energy for residents, respectively. The elastic coefficients of lnEMC and lnEAE were −0.104 and −0.335, respectively, which were significant. Establishing effective energy management institutions and accelerating energy infrastructure development will help improve energy efficiency. In general, with the increasing degree of decentralization, local governments still mainly rely on traditional fossil energy to promote economic development. Nonetheless, the income of residents has also increased in the process, along with the increase in resident access to clean energy. In addition, local governments have improved their views on clean energy and taken some measures that help reduce pollution, such as encouraging the development of energy management agencies, increasing investment in modern energy consumption facilities, and improving resource allocation and energy utilization efficiency.

The total sample is divided into two parts with a higher and a lower degree of economic development by the median per capita GDP to research the nexus between FD and EP at different levels of economic development. The regression consequences are shown in the last two columns of Table 6. In the two subsamples, the elastic coefficients of FD and EP were −0.147 and −0.327, respectively, both of which were significant at the 1% level. The consequences demonstrate that when the economic development level is high, the regression coefficient of FD on EP is larger, meaning FD is greatly beneficial in alleviating EP. In areas with relatively good economic foundations, especially the coastal areas of China, the living standards of people are relatively high. With the support of more sufficient funds, local governments have made greater efforts to popularize clean energy and energy infrastructure has been more complete. In relatively backward, economically underdeveloped areas, however, local governments need to consider more factors, and their primary task may not be energy poverty alleviation but to promote local economic development, so the awareness of the utilization and popularization of renewable energy is lacking.

## 4. Further Analysis

Under the pressure of “race to the top”, local governments will be more attracted to environmental protection issues, and have more incentives to improve energy efficiency and technological innovation, reduce energy costs, and reduce basic energy needs. Alberini and Filippini [46] confirm with a panel dataset of US data that reducing inefficiencies can save around 10% of total energy consumption. Improving energy efficiency is the cheapest and fastest way to meet basic energy needs, allowing residents to consume less energy to achieve the same calorific value, reducing energy demand and thus alleviating energy poverty. Under the fiscal decentralization system, local governments have more initiative and higher power to encourage the development of technological innovation. Technological innovation has advantages in improving energy efficiency, energy cleanliness, and changing the energy consumption structure, which is beneficial in eliminating energy poverty. Technological innovation can not only increase the marginal productivity of energy factors and thus restrain energy demand, but also promote the upgrading of industrial structure, promote the rational distribution of energy factors, realize the intensive utilization of energy factors, and reduce the level of energy consumption.

In the benchmark regression section above, the present study tested the moderate function of TI and EE in the nexus between FD and EP in the form of an interaction term. Here, this research uses the three-step mediation test procedure to further explore the internal impact mechanism of technological innovation and energy efficiency, and set up the following model:(4)$${\mathrm{lnEP}}_{\mathrm{it}}=\delta_{0}+\delta_{1}{\mathrm{lnEP}}_{i,t-1}+\delta_{2}{\mathrm{lnFD}}_{i,t}+\delta_{3}{\mathrm{lnISU}}_{i,t}+\delta_{4}{\mathrm{lnTI}}_{i,t}+\delta_{5}{\mathrm{lnURB}}_{i,t}+\varepsilon_{\mathrm{it}}$$
(5)$${\mathrm{lnEE}}_{\mathrm{it}}=\theta_{0}+\theta_{1}{\mathrm{lnEE}}_{i,t-1}+\theta_{2}{\mathrm{lnFD}}_{i,t}+\theta_{3}{\mathrm{lnISU}}_{i,t}+\theta_{4}{\mathrm{lnTI}}_{i,t}+\theta_{5}{\mathrm{lnURB}}_{i,t}+\varepsilon_{\mathrm{it}}$$
(6)$${\mathrm{lnEP}}_{\mathrm{it}}=\mu_{0}+\mu_{1}{\mathrm{lnEP}}_{i,t-1}+\mu_{2}{\mathrm{lnFD}}_{i,t}+\mu_{3}{\mathrm{lnISU}}_{i,t}+\mu_{4}{\mathrm{lnEE}}_{i,t}+\mu_{5}{\mathrm{lnURB}}_{i,t}+\varepsilon_{\mathrm{it}}$$
(7)$${\mathrm{lnTI}}_{\mathrm{it}}=\varphi_{0}+\varphi_{1}{\mathrm{lnTI}}_{i,t-1}+\varphi_{2}{\mathrm{lnFD}}_{i,t}+\varphi_{3}{\mathrm{lnISU}}_{i,t}+\varphi_{5}{\mathrm{lnEE}}_{i,t}+\varphi_{6}{\mathrm{lnURB}}_{i,t}+\varepsilon_{\mathrm{it}}$$

First, for energy efficiency, Model 4, Model 5, and Model 1 (See Section 2) in the benchmark regression constitute the overall model for examining the indirect effects of energy efficiency. Among them, the coefficient $\delta_{2}$ in Model 4 represents the total effect, and the indirect effect of energy efficiency can only be proved when $\theta_{2}$ in Model 5 and $\alpha_{5}$ in Model 1 are both significant. Additionally, the coefficient $\alpha_{2}$ in Model 1 represents the direct effect. Similarly, Model 6, Model 7, and Model 1 constitute all models for testing the indirect effect of TI, and the explanations of their coefficients are the same as energy efficiency, which will not be repeated here. The regression outcomes for the above models are listed in Table 7.

In Model 4 and Model 6, the coefficient of lnFD is significantly negative at the 1% level, demonstrates that FD is negatively impact EP. The coefficients of lnFD in Model 5 and Model 7 are both positive and significant, indicating that FD can promote the increase in EE and TI. In the last column of Table 7, the lnFD is negatively related to EP, indicating that the direct effect of FD on EP is negative, and the coefficients of lnEE and lnTI are negative and significant. Therefore, the results show that energy efficiency and technological innovation play a mediating role in the nexus between FD and EP, that is, FD increases EE and TI, thereby reducing EP.

## 5. Discussion

The results of the baseline regression show that there is a negative and significant correlation between fiscal decentralization, industrial upgrading, energy efficiency, and technological innovation, which indicate that these factors are helpful to solve the energy problem. However, the results show that the process of urbanization will promote energy poverty.

Fiscal decentralization is a mechanism to regulate the central and local fiscal power, and plays a decisive role in the efficiency and manner of financial resource allocation [21]. On the one hand, the increase in fiscal decentralization may mean that the intensification of degree of game between the local government and the central government, which increases the possibility that the local government violates the central environmental protection policy to promote economic development [47]. On the other hand, the local government may improve the environment quality and increase the use of clean energy through greater fiscal autonomy [22]. Obviously, the results of this paper support the latter. Moderate fiscal decentralization will improve energy efficiency [24], promote the development of renewable energy [21,48], and also have spatial spillover effect [49]. This shows that fiscal decentralization will improve the positive attitude of local governments to solve energy problems.

The adjustment of industrial structure plays an important role in reducing energy intensity [50], which can improve the efficiency of resource allocation and the rationality structure of supply and demand [25]. The upgrading of industrial structure will lead to changes in energy consumption mode, type, and demand, thus alleviating energy poverty. Inefficient energy policies will exacerbate energy poverty [20]. Improving energy efficiency can reduce energy demand and consume less energy at the same calorific value [46]. Energy efficiency is an important measure to improve energy poverty and will further effectively improve the welfare of residents [51]. Technology innovation can improve the energy consumption structure, reduce the energy consumption per unit output, and energy consumption intensity [52], thus have a positive impact on the elimination of energy poverty. Wang et al. [53] and Lee et al. [54] proposed that renewable energy technology innovation has a positive role in the expansion of renewable energy production and the reduction of costs, thus improving the opportunities for residents to obtain and consume renewable energy and helping to address the energy poverty problem. The acceleration of urbanization, on the one hand, will be accompanied by the increase in population; on the other hand, it will increase the demand for urban infrastructure [55], and both of them will increase the use of energy, especially fossil fuel energy [26], which will affect the energy structure and energy, and further aggravate energy poverty.

The results of the heterogeneity test show that fiscal decentralization has an impact on all the four sub-indicators of energy poverty, but the specific effects are different. On the one hand, the decentralization of fiscal power will improve the availability of energy services, energy infrastructure construction, and energy efficiency. The increased autonomy of local governments will enhance the sense of responsibility, strengthen the importance of basic services, and increase the income of residents [56], thus improving the use of energy by residents. On the other hand, the relationship between fiscal decentralization and the cleanliness of energy consumption is positive, which indicates that the local governments promote economic development mainly by consuming fossil energy, which is similar to the conclusion of [47]. Consequences of the mechanism test show that technological innovation and energy efficiency are the paths in fiscal decentralization affecting energy poverty. The results of the comprehensive heterogeneity test and mechanism test show that the local government has increased its awareness of reducing energy poverty after the increase in fiscal autonomy, improved the opportunities available to residents for energy services, and tried to achieve this by improving technological innovation and energy efficiency. However, local governments still insist on using non-clean energy to improve local economic development. In a word, the results of the heterogeneity test and mechanism test comprehensively show that the local government has increased its awareness of reducing energy poverty after increasing its fiscal autonomy and improved the opportunities available to residents for energy services by improving technological innovation and energy efficiency. However, local governments still insist on using non-clean energy to improve local economic development.

## 6. Conclusions 

Using the data from 30 provinces from 2004 to 2017, this paper explores the effects of fiscal decentralization, industrial structure upgrading, energy efficiency, technological innovation, and urbanization on energy poverty.

The study has obtained some interesting results through econometric methods. Firstly, the results in the benchmark regression show that fiscal decentralization can significantly reduce energy poverty, and the relationship between industrial upgrading, energy efficiency, technological innovation, and energy poverty is negatively significant, but urbanization is positively and significantly related to energy poverty. Secondly, this paper examines the moderating effects of energy efficiency and technological innovation in the form of interaction terms. The consequences demonstrate that energy efficiency and technological innovation are essential elements in reducing energy poverty. In addition, the current study divides energy poverty into four sub-indicators, and the results represent that fiscal decentralization is helpful in increasing residents’ access to clean energy and promoting energy management agencies and energy infrastructure. However, fiscal decentralization has not played a role in increasing the cleanliness of energy consumption. In a differentiated economic development level, the impact of fiscal decentralization on energy poverty is different. Finally, this paper explores the mediating effect of energy efficiency and technological innovation using a three-step method, which further confirms the indirect effect of energy efficiency and technological innovation.

Based on the results of this paper, several policy implications are offered for eliminating energy poverty.

Firstly, the focus needs to be on energy-poor areas and implementing targeted energy poverty alleviation policies. Energy poverty and economic poverty are not synchronized, so the poverty situation in each region needs to be accurately identified. The heterogeneity of the development between regions and the geographical environment ought to be noticed, and the matter of energy poverty should be settled according to local conditions. For example, photovoltaic power generation can be developed in areas with sufficient sunlight, and hydropower can be used in areas rich in water resources. To settle the matter of energy poverty thoroughly, a one-size-fits-all policy cannot be adopted. Local governments need to consider local characteristics and advantages to carry out targeted energy poverty alleviation.

Second, delegating rights to lower levels of governments will alleviate environmental degradation and energy poverty. Thus, it is extremely vital to reasonably divide the rights and responsibilities of local and central governments in energy poverty alleviation, further optimizing the fiscal expenditure structure of local government and improving the efficiency of energy poverty alleviation.

Third, according to the research results, energy efficiency and technological innovation are essential elements in reducing energy poverty and also are essential paths to help fiscal decentralization reduce energy poverty, the enlightenment is that we ought to take advantage of the positive role of technological innovation and energy efficiency in energy poverty alleviation, accelerate industrial technology upgrading and reduce energy consumption. In addition, when formulating energy poverty reduction policies, it is not only necessary to vigorously develop the regional economy, optimize energy infrastructure, and enhance energy utilization efficiency, but also to consider the resource shortage caused by the accelerating urbanization process, and replace traditional fossil fuels with clean energy.

The limitations of this paper are as follows. Firstly, only the impact on energy poverty from the aspects of fiscal decentralization, industrial upgrading, energy efficiency, technological innovation, and urbanization are discussed. However, energy poverty has multiple dimensions. In the future, it is necessary to further explore other indicators affecting energy poverty and their impact mechanisms, such as regional energy policies, education level, and living habits of resident. Secondly, limited by the availability of data, only 30 provinces from 2004 to 2017 were selected, and only taking the province as the research unit, the spatial scale is large, which reduces the practical significance of the research conclusions. In the future, the research scale can be reduced to the county or city level to further study the origin and influencing factors of energy poverty.

## Figures and Tables

**Figure 1 ijerph-20-04360-f001:**
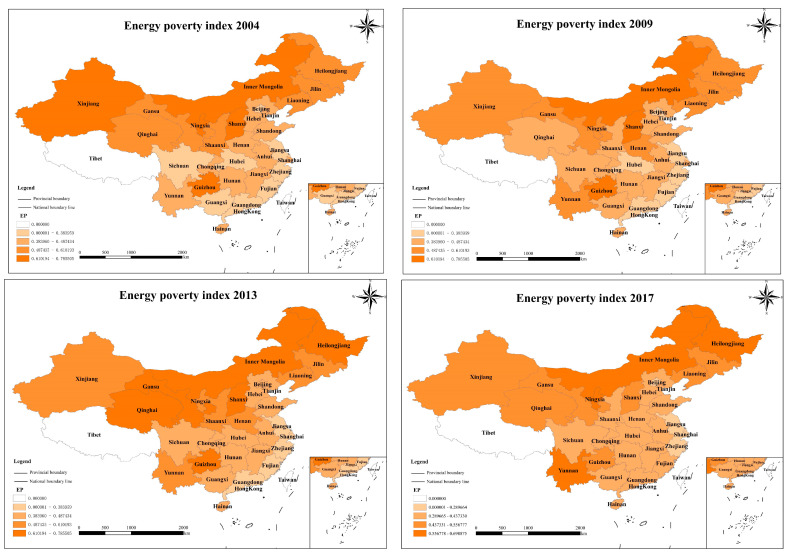
Spatial distribution of energy poverty.

**Figure 2 ijerph-20-04360-f002:**
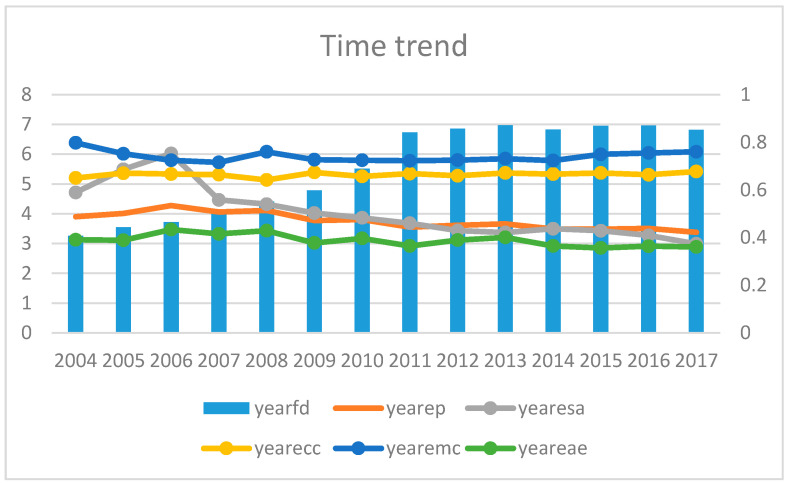
The trends for average energy poverty and average fiscal decentralization.

**Table 1 ijerph-20-04360-t001:** Descriptive statistical analysis of variables.

	(1)	(2)	(3)	(4)	(5)
Variables	N	Mean	Sd	Min	Max
LnEP	420	−0.799	0.296	−1.584	−0.153
LnFD	420	1.582	0.501	0.400	2.697
LnISU	420	−0.107	0.369	−0.704	1.444
LnEE	420	0.047	0.525	−1.464	1.368
LnTI	420	9.081	1.636	4.248	12.710
LnURB	420	−0.689	0.267	−1.974	−0.088

**Table 2 ijerph-20-04360-t002:** EP in distinct region.

Eastern Region		Central Region		Western Region	
Region	EP	Region	EP	Region	EP
Zhejiang	0.2682	Hubei	0.3634	Sichuan	0.3798
Jiangsu	0.3076	Hunan	0.4022	Guangxi	0.3846
Beijing	0.3165	Anhui	0.4157	Chongqing	0.4217
Fujian	0.3199	Jiangxi	0.4258	Shaanxi	0.4894
Guangdong	0.3328	Henan	0.4734	Yunnan	0.5386
Tianjin	0.3460	Jilin	0.5527	Qinghai	0.5504
Shanghai	0.3763	Heilongjiang	0.6649	Gansu	0.5962
Shandong	0.4064	Shanxi	0.6806	Xinjiang	0.6080
Hainan	0.4356			Ningxia	0.6117
Hebei	0.4593			Guizhou	0.6874
Liaoning	0.5667			Inner Mongolia	0.7093
Eastern	0.3759	Central	0.4973	Western	0.5434

**Table 3 ijerph-20-04360-t003:** Cross-sectional dependence test results.

Tests	Statistics	*p*-Value
Breusch–Pagan LM test	1080.77 ***	0.000
Pesaran CD test	13.890 ***	0.000
Friedman test	74.800 ***	0.000

Note: *** <1%.

**Table 4 ijerph-20-04360-t004:** Unit root test results.

Variable	CIPS		CADF	
	Level	First-Difference	Level	First-Difference
LnEP	−1.915	−3.294 ***	−1.932	−2.197 ***
LnFD	−2.104	−3.525 ***	−1.949	−2.150 **
LnISU	−1.868	−3.285 ***	−1.526	−1.985 *
LnEE	−1.811	−3.496 ***	−1.051	−1.743
LnTI	−1.776	−2.671 ***	−1.607	−1.727
LnURB	−1.967	−3.600 ***	−2.195 ***	−2.215 ***

Note: *** < 1%, ** < 5% and * <10%.

**Table 5 ijerph-20-04360-t005:** Results of the impact of FD on EP.

Variables	FGLS			Diff-GMM		
	Model 1	Model 2	Model 3	Model 1	Model 2	Model 3
L.lnEP				0.052 **	0.042 *	0.067 **
				(2.54)	(1.70)	(2.38)
lnFD	−0.188 ***	0.064	−0.177 ***	−0.273 ***	−0.158 ***	−0.276 ***
	(−3.34)	(0.65)	(−3.07)	(−12.00)	(−3.06)	(−14.43)
lnISU	−0.179 ***	−0.168 ***	−0.157 ***	−0.201 ***	−0.177 ***	−0.182 ***
	(−4.46)	(−4.05)	(−3.85)	(−9.37)	(−5.26)	(−6.04)
lnEE	−0.238 ***	−0.216 ***		−0.050 **	−0.019	
	(−4.04)	(−3.49)		(−2.02)	(−0.60)	
lnTI	−0.057 ***		−0.042 **	−0.011 *		−0.004
	(−2.92)		(−2.05)	(−1.67)		(−0.33)
lnURB	0.002	0.009	0.011	0.565 ***	0.515 ***	0.530 ***
	(0.05)	(0.19)	(0.24)	(7.19)	(5.32)	(6.18)
lnFD * lnTI		−0.028 ***			−0.014 **	
		(−3.00)			(−2.53)	
lnFD * lnEE			−0.131 ***			−0.046 ***
			(−4.54)			(−3.10)
Constant	0.311	−0.205	0.139	0.190 **	0.046	0.104
	(1.26)	(−1.43)	(0.52)	(2.38)	(0.58)	(0.76)
Adj-R2	0.134	0.122	0.128			
AR(1)				0.0029	0.0043	0.0015
AR(2)				0.9064	0.9803	0.8570
Sargan test				0.7486	0.9589	0.9558

Note: *** < 1%, ** < 5% and * <10%, ( ) contain the T-values or Z-values.

**Table 6 ijerph-20-04360-t006:** Results of the impact of FD on sub-indexes of EP.

Variables					lnEP	
	lnESA	lnECC	lnEMC	lnEAE	Low	High
L.lnEP					0.168 ***	0.134 ***
					(4.45)	(4.29)
L.lnESA	0.287 ***					
	(47.37)					
L.lnECC		0.219 ***				
		(36.24)				
L.lnEMC			0.043 ***			
			(4.86)			
L.lnEAE				0.166 ***		
				(8.83)		
lnFD	−0.085 **	0.137 ***	−0.104 ***	−0.335 ***	−0.147 ***	−0.327 ***
	(−2.04)	(6.95)	(−10.94)	(−6.13)	(−4.80)	(−6.64)
lnISU	0.055	0.078 ***	0.045 **	−0.224 ***	−0.188 ***	−0.161 ***
	(1.54)	(4.20)	(2.12)	(−7.04)	(−6.99)	(−7.87)
lnEE	−0.118 ***	−0.154 ***	0.225 ***	−0.081	0.029	−0.022
	(−3.22)	(−6.24)	(10.22)	(−1.02)	(0.90)	(−1.28)
lnTI	−0.263 ***	0.012 **	−0.060 ***	0.022	−0.030 ***	−0.023 *
	(−23.23)	(2.28)	(−7.97)	(1.08)	(−3.11)	(−1.71)
lnURB	1.055 ***	−0.117 ***	0.189 **	0.725 *	0.120	0.482 ***
	(4.39)	(−2.61)	(2.36)	(1.77)	(1.12)	(4.46)
Constant	2.687 ***	−0.741 ***	0.538 ***	−0.017	−0.045	0.333 **
	(10.72)	(−14.34)	(5.16)	(−0.04)	(−0.27)	(2.10)
AR(1)	0.0005	0.0139	0.0056	0.0001	0.0042	0.0034
AR(2)	0.0384	0.1433	0.7167	0.3767	0.1798	0.2542
Sargan test	0.9993	0.7571	0.7743	0.9995	0.8764	0.9750

Note: *** <1%, ** < 5%, and * <10%, ( ) contain the Z-values.

**Table 7 ijerph-20-04360-t007:** Results of the mediation analysis.

Variables	EE		TI		
	Model 4	Model 5	Model 6	Model 7	Model 1
L.lnEP	0.056 ***		0.063 ***		0.052 **
	(3.06)		(3.25)		(2.54)
L.lnEE		0.841 ***			
		(67.57)			
L.lnTI				0.393 ***	
				(14.05)	
lnFD	−0.284 ***	0.018 **	−0.284 ***	0.694 ***	−0.273 ***
	(−10.73)	(2.20)	(−11.72)	(19.14)	(−12.00)
lnEE			−0.054 ***	0.630 ***	−0.050 **
			(−2.81)	(9.86)	(−2.02)
lnTI	−0.019 ***	0.053 ***			−0.011 *
	(−2.84)	(17.06)			(−1.67)
lnISU	−0.200 ***	−0.132 ***	−0.198 ***	0.242 ***	−0.201 ***
	(−11.44)	(−9.27)	(−10.33)	(6.34)	(−9.37)
lnURB	0.520 ***	0.006	0.494 ***	0.902 ***	0.565 ***
	(8.27)	(0.17)	(6.56)	(3.94)	(7.19)
Constant	0.238 ***	−0.457 ***	0.058	5.101 ***	0.190 **
	(3.67)	(−10.76)	(0.80)	(14.11)	(2.38)
AR(1)	0.0033	0.0000	0.0020	0.0017	0.0029
AR(2)	0.9660	0.1196	0.8358	0.8279	0.9064
Sargan test	0.7382	1.0000	0.7144	0.9123	0.7486

Note: *** <1%, ** < 5%, and * <10%, ( ) contain the Z-values.

## Data Availability

Some or all data and models that support the findings of this study are available from the corresponding author upon reasonable request.

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
