# Peer review of "Modeling the Impact of Fiscal Decentralization on Energy Poverty: Do Energy Efficiency and Technological Innovation Matter?"

_ijerph, 2023, doi:10.3390/ijerph20054360_

Round 1

Reviewer 1 Report

The paper is of interest to the journal auditorium and offers a novel contribution to the research discussion. There are a few comments to make.

1. In the Introduction the authors offer a description of methods ("Based on the above gaps, this paper examines the impact of FD .... mechanisms"). Since the authors have a separate section on methods and data, I suggest they do not describe methods in the Introduction.

In the Introduction the authors offer a detailed description of their research novelty ("The main contributions of the present study are as follows... policies.). This is too detailed for Introduction, and better fits Conclusion. A more general /theoretic contribution description would fit the Introduction better.

2. There are a few typos/language issues, indicating that the text should be proof-read, i.e. "Under the pressure of "race to the top", local governments will more attractive to environmental protection issues..."(lines 354-355).

3. The first paragraph in Conclusion offers new information and, therefore, belongs to Introduction of Literature review.

Author Response

Response to Reviewer #1:

The paper is of interest to the journal auditorium and offers a novel contribution to the research discussion. There are a few comments to make. (in red)

Response: We appreciate your valuable comments and suggestions. We revised our manuscript following your suggestions. The modifications are highlighted in the paper. Please read our itemized response below. (in red)

Comment 1. Introduction: Please don’t describe methods: -“Based on the above gaps, this paper examines the impact of FD .... mechanisms”.

Introduction: Please simplify the introduction to better fit the conclusion: “The main contributions of the present study are as follows... policies”.

Response 1: Thanks for your valuable suggestions. We reorganized the introduction, deleted the description of measurement methods, and simplified the discussion of research results.    (Please see page 2)

Comment 2. Further discussion: There are a few typos/language issues, indicating that the text should be proof-read, i.e. "Under the pressure of "race to the top", local governments will more attractive to environmental protection issues..."(lines 354-355).

Response 2: Thank you for the kind suggestions. We rechecked the paper and revised the spelling or language problems.

Comment 3. The first paragraph in Conclusion offers new information and, therefore, belongs to Introduction of Literature review.

Response 3: Thank you for the kind suggestions. We have revised this part and added it to the appropriate position in the introduction. (in red)

Reviewer 2 Report

In the article, the authors touched on current and extremely important issues. A complex analysis has been carried out to identify policy implications – so the research is also utilitarian in nature. However, the text needs corrections:

1. The aim of the research should be clearly formulated and included both in the abstract and in the introduction. It is worth supporting the aim with research questions. This is very important because analysis and modeling in themselves cannot be aims, but rather tools.

2. The introduction is too poorly rooted in the literature, there are no references to other studies, 4 references are definitely not enough. In several places it is written that there is a discussion or something missing in the literature, but without references.

3. What does it mean „there is barely any literature on energy poverty from the perspective of fiscal decentralization”? It means that there is something in the literature after all. It is necessary to refer to what is and indicate the research gap. And if there are no articles about China, implement literature from other countries.

4. The literature review is very brief. Authors should consider combining the literature review with the introduction or extending the literature review.

5. Throughout the text, references are incorrectly given. Use a sequential number instead of the year of publication.

6. The discussion should include findings, comparison to the results of other research. As it stands, the discussion looks like a further presentation of the results.

Good luck!

Author Response

Response to Reviewer #2:

In the article, the authors touched on current and extremely important issues. A complex analysis has been carried out to identify policy implications – so the research is also utilitarian in nature. However, the text needs corrections:

Response: We appreciate your valuable comments and suggestions. We revised our manuscript following your suggestions, and the modifications are highlighted in the paper. Please read our itemized response below.(in red)

Comment 1. The aim of the research should be clearly formulated and included both in the abstract and in the introduction. It is worth supporting the aim with research questions. This is very important because analysis and modeling in themselves cannot be aims, but rather tools.

Response 1: Thanks for your valuable suggestions. We revised the abstract and introduction of the paper to clarify the research purpose and described the research issues in more detail.(in red)

Comment 2. Introduction: It is too poorly rooted in the literature, there are no references to other studies, 4 references are definitely not enough. In several places it is written that there is a discussion or something missing in the literature, but without references.

Response 2: Thanks for your valuable suggestions. We supplement the introduction of this study. By integrating the introduction and the literature review, the purpose of improving and enriching the content is achieved.(in red)

Comment 3. What does it mean “there is barely any literature on energy poverty from the perspective of fiscal decentralization”? It means that there is something in the literature after all. It is necessary to refer to what is and indicate the research gap. And if there are no articles about China, implement literature from other countries.

Response 3: Thank you for your valuable comment. We apologize for the unclear description and have revised it in the introduction to improve its clarity. (in red)

Comment 4. The literature review is very brief. Authors should consider combining the literature review with the introduction or extending the literature review.

Response 4: Thank you for your valuable suggestions. We have revised the introduction by integrating it with the literature review, and we have also supplemented the literature review.(in red)

Comment 5.  Throughout the text, references are incorrectly given. Use a sequential number instead of the year of publication.

Response 5: We apologize for the inconsistency in the format of the reference documents. We have carefully revised them to comply with the journal guidelines.(in red)

Comment 6. The discussion should include findings, comparison to the results of other research. As it stands, the discussion looks like a further presentation of the results.

Response 6: Thank you for your valuable comments. We agree with your suggestions and have therefore included a new chapter titled 'Discussion' to further elaborate on our results. We have also enhanced the discussion of results by comparing our findings with existing research.(in red)

Reviewer 3 Report

I reviewed the manuscript with great interest, and have a few minor comments for the authors:

1.       The abstract is lacking the sampling technique and data source. Also, the last sentence is very general “Finally, various policy suggestions for eradicating energy poverty are proposed”. The suggestion should be very specific based on current findings.

2.       The authors have substantially used the word “We”, and “Our”. It is advised to avoid such non-academic words and replace them with academic words, such as this research, the current study, etc.

3.       In the introduction, page 2 and paragraph 2: “We relied on the Generalized Method of Moments (Diff-GMM), and Feasible…………………”. This paragraph should be moved to the methodology section.

4.       The introduction section lacks a discussion about the significance of the study. The authors have to add arguments with valid references that how fiscal decentralization, industrial structure upgrading, energy efficiency, technological innovation, and urbanization affect energy poverty.

5.       The authors have used different citation styles, somewhere they used APA format and somewhere they used IEEE format. It should be the same throughout the draft.

6.       Most of the literature is outdated. This section should develop using the most recent literature related to the studied problem.

7.       In the theoretical framework, I have not found any discussion of underpinning theory. Which theory(s) has to be used to support the current model? It is also advised to develop the research hypothesis with the support of relevant theory before formulating the econometric models.

8.       What is the date source? And which sampling method was used for data selection? This is even not discussed in the methodology section.

9.       The results section is explained and interpreted properly, however, there is no discussion of current findings. It is advised to add a separate section of “Discussion” before the conclusion section.

10.   Lastly, add the limitations at the end in a separate section. 

Author Response

Response to Reviewer #3:

I reviewed the manuscript with great interest, and have a few minor comments for the authors:

Comment 1. The abstract is lacking the sampling technique and data source. Also, the last sentence is very general “Finally, various policy suggestions for eradicating energy poverty are proposed”. The suggestion should be very specific based on current findings.

Response 1: Thank you for your valuable suggestions. We have provide the policy implications in the abstract with concise sentences. However, a detailed discussion of the policy implications is provided in a separate section after the conclusion.(in red)

Comment 2. The authors have substantially used the word “We”, and “Our”. It is advised to avoid such non-academic words and replace them with academic words, such as this research, the current study, etc.

Response 2: Thanks for your valuable suggestions. We are sorry to use many non-academic words, which have been revised now.(in red)

Comment 3.  In the introduction, page 2 and paragraph 2: “We relied on the Generalized Method of Moments (Diff-GMM), and Feasible…………………”. This paragraph should be moved to the methodology section.

Response 3: Thanks for your valuable suggestions. We deleted this describe of method in the introduction and interpreted the methods used in ‘3.3 Econometric methods’.(in red)

Comment 4. The introduction section lacks a discussion about the significance of the study. The authors have to add arguments with valid references that how fiscal decentralization, industrial structure upgrading, energy efficiency, technological innovation, and urbanization affect energy poverty.

Response 4: Thanks for your valuable suggestions. In introduction section, We reorganized and supplemented the research significance, and made a general summary of the relationship between the variables used in this paper. The specific analysis of the relationship between the variables is presented in the "empirical results" and "discussion sections".(in red)

Comment 5. The authors have used different citation styles, somewhere they used APA format and somewhere they used IEEE format. It should be the same throughout the draft.

Response 5: Thanks for your valuable suggestions. We are very sorry for this mistake, and we have changed the format of the reference.(in red)

Comment 6. Most of the literature is outdated. This section should develop using the most recent literature related to the studied problem.

Response 6: Thanks for your valuable suggestions. We have supplemented and replaced the references. (in red)

Comment 7. In the theoretical framework, I have not found any discussion of underpinning theory. Which theory(s) has to be used to support the current model? It is also advised to develop the research hypothesis with the support of relevant theory before formulating the econometric models.

Response 7: Thank you for taking the time to review our manuscript and for providing valuable feedback. We appreciate your suggestion regarding the theoretical framework of our study. We acknowledge the lack of discussion on the underpinning theory related to fiscal decentralization and energy poverty in our manuscript. After careful consideration, we have incorporated theory theories related to fiscal decentralization and energy poverty: the theory of fiscal federalism suggests that decentralization of fiscal power can promote efficient and effective public service delivery at the local level by creating incentives for local governments to improve their own revenue base and expenditure decisions [27]. This theory implies that fiscal decentralization may positively impact energy poverty alleviation by empowering local governments to develop and implement policies and programs that target the needs of energy-poor households. These theories are integrated into our research model, which includes hypotheses developed with their support, and the econometric models are formulated accordingly.(in red)

Comment 8. What is the date source? And which sampling method was used for data selection? This is even not discussed in the methodology section.

Response 8: Thank you for your valuable feedback. We apologize for the oversight in not explicitly mentioning the date source and sampling method used in our methodology section. We utilized data from the China Financial Statistics Yearbook for fiscal decentralization, while the data on Industrial Structure Updating, Technology Innovation, and Urbanization are obtained from China Statistics Yearbook, while data of energy efficiency is obtained from Energy Statistics Yearbook, and China Statistics Yearbook. At present, no uniform measure of energy poverty has been derived, and this paper draws on Wang et al. [9] to classify EP into four dimensions: energy service availability (ESA), energy consumption cleanliness (ECC), energy management integrity (EMC), household energy affordability and Energy Efficiency (EAE), calculated following improved entropy method (IEM) by 17 indicators. 

In order to study the effect of FD on EP, a balanced panel of 30 Chinese provinces from 2004-2017 is used in this study. Considering the data availability, the sample data of Tibet, Hong Kong, Macau, and Taiwan are not covered yet. The starting period of 2004 is based on FD data, and the ending period of 2017 is linked with data availability for EP. (in red)

Comment 9. The results section is explained and interpreted properly, however, there is no discussion of current findings. It is advised to add a separate section of “Discussion” before the conclusion section.

Response 9: Thanks for your valuable suggestions. We added a new section "Discussion" before the "conclusion", and strengthen the literature dialogue by comparing with the existing literature conclusions.(in red)

Comment 10. Lastly, add the limitations at the end in a separate section.

Response 10: Thanks for your valuable suggestions. We have recombined the structure of the conclusion part and made the limitation as a separate part.(in red)

Round 2

Reviewer 2 Report

I would like to thank the Authors for taking into account my suggestions and answers.

Reviewer 3 Report

The authors have incorporated the suggest comments. And the revised daft is more reader friendly.